# Sperm-Associated Antigen 5 Knockout Reduces Doxorubicin and Docetaxel Resistance in Triple-Negative Breast Cancer MDA-MB-231 and BT549 Cells

**DOI:** 10.3390/cancers16071269

**Published:** 2024-03-24

**Authors:** Ji He, Jiawei Li, Yanbiao Liu, Yan Li

**Affiliations:** 1School of Science, Auckland University of Technology, Auckland 1010, New Zealand; ji.he@aut.ac.nz (J.H.); lijiawei1990@sina.com (J.L.); gww8665@autuni.ac.nz (Y.L.); 2Department of Food and Agriculture Technology, Yangtze Delta Region Institute of Tsinghua University, Jiaxing 314006, China; 3General Medicine Department, Shenzhen Longhua District Central Hospital, Shenzhen 518110, China

**Keywords:** SPAG5, Astrin, triple-negative breast cancer, CRISPR-Cas9, knockout, doxorubicin, docetaxel

## Abstract

**Simple Summary:**

Sperm-associated antigen 5 (SPAG5) is associated with tumour initiation, progression, and resistance to front-line therapeutics in many cancer types. However, the role of SPAG5 in triple-negative breast cancer (TNBC) remains controversial, and evidence for SPAG5-mediated chemoresistance in TNBC is lacking. Our research demonstrates that SPAG5 may be an important determinant of the efficacy of doxorubicin and docetaxel, the most widely used chemotherapy, in TNBC. We focused on the genetic removal of the *SPAG5* gene by using a gene editing technique to determine if the SPAG5 deletion in the TNBC cell lines could downregulate chemoresistance. We found that the SPAG5 deletion contributed to a sensitive phenotype in the TNBC cells. Our results may help practitioners predict patients’ response to doxorubicin and docetaxel, and targeting SPAG5 may represent a promising way to overcome chemoresistance in TNBC patients.

**Abstract:**

Sperm-associated antigen 5 (SPAG5), also known as Astrin, was previously demonstrated as a biomarker for cellular resistance to major breast cancer therapies, including chemo-, endocrine- and targeted therapy. However, the contribution of SPAG5 to anthracycline- and taxane-based chemotherapy in triple-negative breast cancer (TNBC) remains controversial. In the present study, the *SPAG5* knockout cell model was established by using clustered regularly interspaced palindromic repeats (CRISPR)-CRISPR-associated protein 9 (Cas9) system in MDA-MB-231 and BT549 TNBC cell lines. The knockout of SPAG5 was confirmed on both gene and protein levels using genomic PCR, DNA sequencing and western blotting. The functional loss of SPAG5 was determined by colony-formation assay. SPAG5-regulated doxorubicin- and docetaxel-resistance was assessed by MTT and apoptosis assays. The results indicated that all the *SPAG5* knockout MDA-MB-231 and BT549 clones were biallelic, where one allele was replaced by the donor template, and the other allele had the same “T” insertion (indel) adjacent to the cutting sites of gRNAs at the exon 1 boundary, irrespective of the gRNAs and cell lines. The locus of indel interrupted the *SPAG5* transcription by damaging the GT-AG mRNA processing rule. Deletion of SPAG5 decreased clonogenicity in both MDA-MB-231 and BT549 cells. SPAG5 was able to regulate the resistance and the drug-induced apoptosis of both doxorubicin and docetaxel. In conclusion, recombinant plasmid-based CRISPR-Cas9 technology can be used to delete the *SPAG5* gene in the TNBC cell lines. SPAG5 has an important role in regulating cell proliferation and doxorubicin- and docetaxel-resistance in MDA-MB-231 and BT549 cells.

## 1. Introduction

SPAG5 (aka., Astrin or hMAP126) is a nuclear protein that is encoded by the *SPAG5* gene on Ch17q11.2, the region in which human epidermal growth factor receptor 2 (HER-2)/neu proto-oncogene and peroxisome proliferator-activated receptor binding protein (PBP/PPARBP) are amplified. SPAG5 was first cloned in Hela cells in 2001 as a mitotic spindle-associated protein, which is a regulator of mitotic spindle formation and chromosome segregation [1,2]. SPAG5 contributes to kinetochore–microtubule attachments, and the depletion of this protein results in a spindle checkpoint arrest, premature centriole disengagement, and a loss of sister chromatid cohesion [3]. Recent studies have found that *SPAG5* is a potential driver oncogene in a wide spectrum of cancers including breast cancer, non-small cell lung cancer, prostate cancer, cervical cancer, and bladder urothelial cancer [4,5,6,7,8,9]. An upregulated expression of SPAG5 is correlated with poor clinical outcomes and adverse clinicopathological features. The prevalent gene gains and amplifications of *SPAG5* facilitate neoplastic growth, chemoresistance, metastasis, local recurrence, and overall survival (OS), whereas downregulated SPAG5 expression in several cancer cell models impairs cell proliferation and migration motility and resensitises cells to anticancer therapy [4,5,6,7,8,9]. SPAG5 is thought to regulate various signalling pathways in connection to tumour initiation and progression, such as the hyperactivation of the mammalian target of rapamycin (mTOR) complex 1 (mTORC1), the p53-mediated DNA damage response, phosphatidylinositol 3-kinase/protein kinase B/mTOR (PI3K/AKT/mTOR), AKT/mTOR/WNT, and the activation of c-MYC pathways [10,11,12,13,14]. In addition, a comprehensive analysis of tumour immunity for cancer immunotherapy has also identified SPAG5 as a potential candidate for future vaccine development in multiple cancers, in which the median SPAG5 expression in these cancers is at least threefold over normal [15]. Thus, SPAG5 could serve as a biomarker in predicting cancer proliferation, progression, and response to therapy and clinical outcomes. Targeting SPAG5 may represent a novel strategy to overcome barriers to current cancer therapies such as metastasis or recurrence after surgery and resistance to standard chemotherapy, endocrine therapy, and targeted therapies.

Although SPAG5 has pleiotropic roles in many cancer types, the function of SPAG5 in TNBC is not well defined and understood. Previous artificial neural network-based data mining research suggested that the transcript and protein products of *SPAG5* were independent predictors for response to combination chemotherapy in oestrogen receptor (ER)-negative (ER-) breast cancer [9]. However, another study argued that the higher SPAG5 expression was not associated with an increased risk of relapse and shorter OS in ER-breast cancer. The discrepancy between these observations could be derived from the different chemotherapy regimens and methodologies [16]. Given the controversial findings, the predictive cutoff values and the drug-targeting potential of SPAG5 in TNBC merit further investigation.

However, previous research into the function of SPAG5 in breast cancer was mainly based on the traditional gene knockdown cell models [13,17,18,19]. The incomplete protein depletion and the relatively high false positive rate induced by off-target effects frequently complicate the interpretation and applicability of the findings [20,21]. In recent years, more and more attention has been drawn to CRISPR-Cas9 gene knockout technology. In comparison with gene knockdown technologies (e.g., RNA interference), CRISPR-Cas9 is superior considering its low off-target activity, constant and heritable deletion of the target gene, and multi-target ability [21,22,23]. Our previous review summarised and discussed the current CRISPR-Cas systems used in mammalian genomic editing and the mechanisms and limitations underlying CRISPR-Cas9 [21]. In the present study, SPAG5 knockout TNBC cells were established by using CRISPR/Cas9 technology. To the best of our knowledge, this is the first CRISPR-mediated SPAG5 knockout TNBC cell model worldwide. We give a new perspective on the target localisation of CRISPR-Cas9 and the repair pathways involved. Additionally, our findings first provided direct evidence of the contribution of SPAG5 to TNBC chemoresistance. Thus, this study has significant instructions for the application of CRISPR-Cas9 in studying genetic determinants of TNBC. Our results may provide a novel strategy for stratifying TNBC patients for precision regimens and future perspectives on its clinical application.

In this study, we hypothesised that SPAG5 is a regulator of TNBC development and chemoresistance. Recombinant plasmid-based CRISPR-Cas9 technology was employed to develop monoclonal *SPAG5* gene knockouts in MDA-MB-231 and BT549 TNBC cell lines. Deletion of *SPAG5* gene expression in the selected knockout clones was determined by genomic PCR, DNA sequencing and Western blotting. The role of SPAG5 in cancer hallmarks of TNBC cells was investigated by a colony formation assay; and the effects of SPAG5 on the chemoresistance of doxorubicin and docetaxel was investigated by cytotoxicity assay, caspase 3/7 activity analysis, mitochondrial membrane potential assay, flow cytometry, and fluorescence microscopy.

## 2. Materials and Methods

### 2.1. Cell Culture

The human TNBC cell lines, MDA-MB-231 and BT549, were from the American Type Culture Collection (ATCC, Manassas, VA, USA). MDA-MB-231 and BT549 cell lines were cultured using complete DMEM medium (cat. no. 11965-092, Life Technology, Auckland, New Zealand) supplemented with 10% (*v*/*v*) FBS (cat. no. MG-FBS0820, MediRay, Auckland, New Zealand) and 2 mmol/L L-glutamine (cat. no. 20530081, Life Technology, New Zealand) in a humidified atmosphere of 5% carbon dioxide at 37 °C.

### 2.2. Transfection and Isolation of SPAG5 Gene Knockout Clones

Cells were seeded into a 6-well plate at a density of 3 × 10^5^ cells/well (nearly 70% confluence) in complete medium 24-h prior to transfection. The plasmid encoding guide RNA (gRNA) sequence (pCasG-SPAG5-gRNA1 or pCasG-SPAG5-gRNA2, OriGene Technology, Inc., Rockville, MD, USA), donor plasmid (pUC-SPAG5 donor Luciferase-puro, OriGene Technology, Inc.) and TurboFect Transfection Reagent (cat. no. R0531, ThermoFisher Scientific, Auckland, New Zealand) were gently mixed with Opti-MEM^TM^ I reduced serum medium (cat. no. 31985070, ThermoFisher Scientific, New Zealand) in a fresh and RNAse-free microcentrifuge tube according to the manufacturer’s instructions (OriGene Technology, Inc.). The target sequence was “5′-CCTTCGCCCCAGACGGTAAG-3′” for gRNA1 and “5′-AGATCTCCCGCTTACCGTCT-3′” for gRNA2. The transfection complex was mixed thoroughly, incubated for 15 min at room temperature, and then added to the adherent cells. After 36-h incubation, the transfection complex was removed. In order to isolate monoclonal knockout clones, the pooled transfected cells were immediately dispensed into eight Petri dishes with fresh complete medium containing 0.4% puromycin (cat. no. A1113803, ThermoFisher Scientific, New Zealand) to dilute out wild-type (WT) cells. The puromycin-containing medium was changed every 2 to 3 days until single-cell colonies could be observed (2–4 weeks). The colonies were picked up and seeded in 96-well plates with the aid of pipettor tips and a microscope.

### 2.3. DNA Extraction, PCR and Sequencing

To screen the single-cell clones, the genomic DNA was extracted and purified using the QIAamp DNA Mini Kit (cat. no. 51306, QIAGEN, Hilden, Germany) according to the manufacturer’s instructions. The total DNA was quantified using a NanoDrop^®^ ND-1000 UV spectrophotometer (NanoDrop Technologies, Wilmington, DE, USA). An aliquot of extracts then went through either PCR amplification using TaKaRa PCR Amplification Kit (cat. no. R011, TaKaRa Bio, Inc., San Jose, CA, USA) or Sanger Sequencing (Massey Genome Service, Massey University, Palmerston North, New Zealand) according to the manufacturer’s instructions. The primer constructs listed in Table 1 were purchased from ThermoFisher Scientific (ThermoFisher Scientific, New Zealand). To verify the length of the amplicon, agarose gel analysis was employed to detect and visualise the PCR products.

### 2.4. Western Blotting

SPAG5 protein expression was confirmed by western blotting. Cells grown in a T25 flask (90% confluence) were washed with ice-cold PBS and lysed with 1 mL of modified Laemmli lysis buffer (62.5 mM tris-HCL (pH 6.8), 2% SDS, 10% glycerol, 100 mM dithiothreitol, and 0.01% bromophenol blue) [24] for 10 min at room temperature with agitation. The cell lysate was then passed through a 27-gauge needle at least 10 times to reduce viscosity. An aliquot of 45 μL of cell lysate was loaded in Mini-PROTEAN^®^ TGX^TM^ Precast Gel (10%) (cat. no. 4561034, BIO-RAD, Auckland, New Zealand) and proceeded to gel electrophoresis. The separated proteins were transferred onto a polyvinylidene difluoride (PVDF) membrane (cat. no. 1704156, BIO-RAD, New Zealand) using Trans-Blot^®^ Turbo^TM^ Blotting System (Bio-Rad Laboratories, Inc.). The PVDF membrane was blocked with blocking buffer (filtered 2% BSA in TBST buffer) for 1 h at room temperature with agitation. The membrane was then probed by SPAG5 Mouse Monoclonal Antibody (Clone ID: OTI3F10) (cat. no. TA810452, OriGene Technology, Inc.) overnight at 4 °C and Anti-Mouse IgG (whole molecule)—Peroxidase antibody produced in rabbit (cat. no. A9044, Sigma-Aldrich, Auckland, New Zealand) for 1 h at room temperature with agitation and visualised by Clarity^TM^ Western ECL Substrate (cat. no. 1705060, BIO-RAD, New Zealand). The chemiluminescence was detected using ImageQuant^TM^ LAS 500 (Bio-Rad Laboratories, Inc.). For beta-actin control, the blots on the PVDF membrane were stripped off and re-stained with Monoclonal Anti-β-Actin Antibody Produced in Mouse (cat. no. A2228, Sigma-Aldrich, New Zealand) and Anti-Mouse IgG (whole molecule)—Peroxidase antibody produced in rabbit (cat. no. A9044, Sigma-Aldrich, New Zealand).

### 2.5. Colony-Formation Assay

Cells were plated in 6-well plates at a density of 200 cells/well. The cells were allowed to grow for 10 to 20 days at 37 °C and 5% CO_2_ until clear cell colonies were observed. The old culture medium was replaced with a fresh complete medium every 2 to 3 days during the incubation period. After incubation, cells were fixed with 4% paraformaldehyde (PFA) for 15 min at room temperature, followed by a 15 min-incubation with 0.5% crystal violet. The stained cells were gently rinsed with tap water and air-dried. The plates were imaged and analysed under an inverted microscope (Zeiss, Oberkochen, Germany). Colonies of more than 50 cells are scored for survival and counted using ImageJ (1.50i) software.

### 2.6. Cytotoxicity Assay

Cells were plated at a density of 5000 cells/well in 96-well plates and allowed to attach for 24 h. After incubation, cells were exposed to either doxorubicin (Doxorubicin Ebewe^®^ Hydrochloride Injection, Sandoz Pty Ltd., North Sydney, Australia) or docetaxel (DBL^TM^ Docetaxel Concentrated Injection, Pfizer Australia Pty Ltd., Sydney, Australia) at various concentrations for 72 h. After treatment, the drug-containing medium was removed, and 100 μL of fresh complete medium was added to each well to terminate drug exposure. An aliquot of 10 μL of 12 mM 3-(4,5-dimethylthiazol-2-yl)-2,5-diphenyltetrazolium bromide (MTT) (cat. no. M2128, Sigma-Aldrich, Auckland, New Zealand) solution was then added to the cells and incubated for 4 h at 37 °C, followed by addition of 150 μL of DMSO. The cell viability was quantified by measuring absorbance at 570 nm. Data were normalised to the mean absorbance of vehicle control. The IC_50_ values were assessed using nonlinear regression in GraphPad Prism 7.0 software (GraphPad Software, Inc., La Jolla, CA, USA).

### 2.7. Caspase 3/7 Activity Analysis

The caspase3/7 activity in MDA-MB-231 and BT549 cells treated with doxorubicin was determined by using Caspase-Glo^®^ 3/7 Assay kit (cat. no. G8091; Promega Corporation, Madison, WI, USA) considering the autofluorescence of doxorubicin. Caspase-Glo^®^ 3/7 Reagent generates luminescent signal, which is proportional to caspase 3/7 activity in the tested cells. Cells were seeded at a density of 5000 cells per well in Nunc^TM^ MicroWell^TM^ 96-Well, Nunclon Delta-Treated, Flat-Bottom Microplates (cat. no. 136101; Thermo Fisher Scientific, Inc.) and incubated overnight at 37 °C and 5% CO_2_. MDA-MB-231 and BT549 cells were then treated with different concentrations of doxorubicin. After 72-h incubation, the drug-containing medium was replaced with 100 μL fresh medium. An aliquot of 100 μL of Caspase-Glo^®^ 3/7 Reagent was then added to each well and gently mixed using a plate shaker at 400 rpm for 30 s, followed by a 1-h incubation at room temperature in the dark. The luminescence was measured using a SPARK^®^ Microplate Spectrofluorometer (Molecular Devices, LLC, San Jose, CA, USA).

The docetaxel-induced caspase 3/7 activity changes in MDA-MB-231 and BT549 cells were assessed by using CellEvent™ Caspase-3/7 Detection Reagent (cat. no. C10427; Thermo Fisher Scientific, Inc.). Cells were plated in 96-well plates at a density of 5000 cells/well. After overnight incubation at 37 °C and 5% CO_2_, MDA-MB-231 and BT549 cells were exposed to various concentrations of docetaxel for 72 h. After exposure, the drug-containing medium was removed, and the cells were then soaked in 100 μL of diluted CellEvent™ Caspase-3/7 Detection Reagent (final concentration 2 μM) for 60 min. Data were collected on a SPARK^®^ Microplate Spectrofluorometer (Molecular Devices, LLC) by measuring the fluorescence at ex 467/em 539 nm. The cells were also imaged using a Leica DMi8 inverted fluorescence microscope (Leica Microsystems, GmbH, Wetzlar, Germany).

### 2.8. Apoptosis Detection

Dead Cell Apoptosis Kit with Annexin V Alexa Fluor™ 488 and Propidium Iodide (PI) (cat. no. V13245, ThermoFisher Scientific, New Zealand) was used to further confirm the apoptotic effect induced by docetaxel in MDA-MB-231 and BT549 cells. Cells were seeded at a density of 2 × 10^5^ cells per well in 6-well plates 24-h prior to docetaxel treatment. After 72 h exposure to docetaxel, the cells were harvested and double-stained with annexin V and PI according to the manufacturer’s instructions. The fluorescence was measured at ex 530/em 575 nm using MoFlo^TM^ XDP flow cytometer (Beckman Coulter, Inc., Brea, CA, USA) and analysed with Kaluza software (version 1.3) (Beckman Coulter, Inc., CA, USA).

### 2.9. Mitochondrial Membrane Potential Assay

The mitochondrial membrane potential of MDA-MB-231 cells was measured using TMRE-Mitochondrial Membrane Potential Assay Kit (cat. no. ab113852, Abcam, Nelson, New Zealand). Cells were seeded at a density of 2 × 10^5^ cells per well in 6-well plates and incubated overnight at 37 °C and 5% CO_2_. The cells were treated with 0-, 1- and 3-nM docetaxel for 72-h, respectively. The cells were then harvested and incubated with TMRE for 30 min. After incubation, the cells were washed with PBS containing 0.2% BSA. The fluorescence was measured at ex 488/em 575 nm using MoFlo^TM^ XDP flow cytometer (Beckman Coulter, Inc., CA, USA) and analysed with Kaluza software (version 1.3) (Beckman Coulter, Inc., CA, USA).

### 2.10. Statistical Analysis

The statistical analysis was carried out using GraphPad PRISM8 (GraphPad; Dotmatics). The difference between various groups was analysed using either one-way or two-way analysis of variance (ANOVA) with appropriate post-hoc tests and reported as the mean ± standard deviation. A cutoff value of *p* = 0.05 was applied, where *p* < 0.05 was considered statistically significant.

## 3. Results

### 3.1. Deletion of SPAG5 Expression in SPAG5 Knockout Clones

Recombinant plasmid-based CRISPR-Cas9 technology was employed to develop monoclonal *SPAG5* gene knockouts in MDA-MB-231 and BT549 cell lines. Cells were transfected with plasmid donors (pUC-SPAG5 donor Luciferase-puro) and plasmids encoding Cas9 protein and the gRNAs (pCasG-SPAG5-gRNA1 or pCasG-SPAG5-gRNA2) to replace *SPAG5* exon 1 and 2 with donor template, followed by clone selection. The isolated clones were screened by chromosome-specific PCR that examines the homology-directed repair (HDR)-mediated on-target integration of donor sequence (Appendix A). The positive clones were further tested for homozygosity using a pair of primers flanking the cutting site of gRNA1 and gRNA2, which hybridise the target *SPAG5* genomic sequence. Unfortunately, none of the positive clones were with bi-allelic integration of donor sequence (Appendix A). The allele without donor sequence in the heterozygous knockout clones might be cleaved and repaired by a nonhomologous end-joining (NHEJ) DNA repair pathway, which means that one allele was replaced by the donor sequence and the other allele might be either intact or indel. Thus, the PCR products were further sequenced. Interestingly, the indel allele of different clones had the same “T” insertion at the same locus, 3 bp downstream from *SPAG5* exon 1, irrespective of the gRNAs and cell lines (Appendix A). The locus of this “T” insertion on the indel allele potentially interrupts the *SPAG5* transcription and, thus, protein expression by damaging the conformation of intron–exon boundaries (GT-AG rule) [25].

One of each gRNA1 and gRNA2 positive clones was adopted and proceeded to downstream assays to eliminate single-cell variability, namely clone MDA-MB-231-gRNA1 (M c1), clone MDA-MB-231-gRNA2 (M c2), clone BT549-gRNA1 (B c1) and clone BT549-gRNA2 (B c2). *SPAG5* gene editing in the selected clones was examined by genomic PCR, DNA sequencing and Western blotting. Chromosome-specific PCR demonstrated that the selected clones detected on-target integration of both the puromycin gene (Figure 1A-I) and luciferase gene (Figure 1A-II) encoded by donor sequence. Western blotting analysis of the SPAG5 expression in the selected clones identified the complete deletion of the SPAG5 protein (Figure 1B). DNA sequencing showed the “T” insertion on the indel allele of the selected clones (Figure 1A-III,C). 

SPAG5 has been implicated in cell proliferation and migration motility in breast cancer [10]. To further determine the functional removal of the SPAG5 protein, the clonogenic capacity of the selected clones was examined using a colony-formation assay. Knockout of *SPAG5* decreased the clonogenic capacity of MDA-MB-231 and BT549 cells. The colonies formed within the incubation period for the WT control MDA-MB-231 and BT549 cells were significantly more numerous than those of the *SPAG5* knockout clones (Figure 2).

### 3.2. SPAG5 Knockout Sensitised TNBC Cells to Doxorubicin- and Docetaxel-Induced Growth Inhibition

MDA-MB-231 and BT549 clones were treated with 0–1000 nM doxorubicin and 0–25 nM docetaxel, respectively. The 72-h cytotoxicity of doxorubicin and docetaxel was determined using an MTT assay.

Both doxorubicin and docetaxel inhibited the proliferation of WT control and knockout clones in a dose-dependent manner (Figure 3). Table 2 shows that the IC_50_ values were decreased by 59% (M c1) and 48% (M c2) for doxorubicin and 52% (M c1) and 45% (M c2) for docetaxel in MDA-MB-231 knockout clones compared to the WT control, respectively. A similar trend was found in BT549 knockout clones; the IC_50_ values in B c1 and B c2 exhibited a reduction of 38% and 72% for doxorubicin and a reduction of 25% and 27% for docetaxel, respectively (Table 3).

### 3.3. SPAG5 Knockout Increased Doxorubicin- and Docetaxel-Induced Apoptosis Rate

Given the autofluorescence of doxorubicin, the Caspase-Glo^®^ 3/7 Assay kit generating luminescent signal was adopted to detect the caspase 3/7 activity in MDA-MB-231 and BT549 cells treated with doxorubicin. The time course remained the same as the MTT assay, and the drug concentration was adjusted according to the IC50 values determined in the MTT assay. Figure 4 shows that the caspase 3/7 activity was significantly higher in *SPAG5* knockout clones than in the WT control after exposure to different concentrations of doxorubicin. A doxorubicin concentration of 20 nM induced caspase 3/7 activity in MDA-MB-231 knockout clones M c1 and M c2 increased by 18.06 ± 4.5% (*p* < 0.01) and 16.84 ± 4.2% (*p* < 0.01) compared to the WT control, respectively (Figure 4A). When treated with 70 nM doxorubicin, M c1 and M c2 had caspase 3/7 activity increased by 19.4 ± 6.57% (*p* < 0.001) and 14.51 ± 3.96% (*p* < 0.05) compared to the WT control, respectively (Figure 4A). Similarly, BT549 knockout clones B c1 and B c2 exhibited an increase of 14.14 ± 7.45% (*p* < 0.05) and 20.4 ± 3.96% (*p* < 0.001) in caspase 3/7 activity with treatment of 3 nM doxorubicin and an increase of 14.58 ± 3.19% (*p* < 0.01) and 23.61 ± 2.28% (*p* < 0.0001) with 10 nM doxorubicin, respectively.

Docetaxel-induced caspase 3/7 activity changes were measured using fluorescent CellEvent^TM^ Caspase-3/7 Green Detection Reagent. Both MDA-MB-231 and BT549 knockout clones exposed to docetaxel exhibited enhanced caspase3/7 activity than the WT control after 72-h incubation. In MDA-MB-231 clones M c1 and M c2 treated with 1 nM docetaxel, the caspase 3/7 activity increased by 47.88 ± 7.98% (*p* < 0.0001) and 38.72 ± 8.06% (*p* < 0.01) compared to the WT control, respectively (Figure 5A,B). After treatment with 3 nM docetaxel, the caspase 3/7 activity increased by 26.41 ± 2.9% (*p* < 0.05) and 25.81 ± 6.52% (*p* < 0.05) for M c1 and M c2, respectively (Figure 5A,B). A similar trend was seen in BT549 cells—the caspase 3/7 activity increased by 31.88 ± 2.9% (*p* < 0.001) and 22.02 ± 5.5% (*p* < 0.01) in B c1, and 40.89 ± 2.76% (*p* < 0.0001) and 25.85 ± 8.3% (*p* < 0.001) in B c2, after treatment with 0.4 nM and 0.6 nM docetaxel compared to the WT control, respectively (Figure 5C,D).

The apoptotic effects of docetaxel in the TNBC cells were further confirmed by using the Dead Cell Apoptosis Kit with Annexin V Alexa Fluor™ 488 and PI. Both MDA-MB-231 and BT549 cells underwent apoptosis in a dose-dependent manner. The apoptosis rate was significantly elevated in knockout clones compared to that in the WT control. The apoptosis rate in MDA-MB-231 knockout clones M c1 and M c2 with 1 nM docetaxel treatment was 8.31 ± 1.22% (*p* < 0.01) and 7.63 ± 0.35% (*p* < 0.01) compared to that of 4.09 ± 1.88% in the WT control, respectively (Figure 6A,B). The difference in the apoptosis rate between knockout clones and the WT control widened further when treated with 3 nM docetaxel. The apoptosis rate in M c1 and M c2 with 3 nM docetaxel treatment was 12.55 ± 2.39% (*p* < 0.0001) and 11.04 ± 2.45% (*p* < 0.0001) compared to that of 4.8 ± 1.87% in the WT control, respectively (Figure 6A,B). Similarly, the apoptosis rate in BT549 knockout clones with 1nM and 3nM docetaxel treatment was 12.24 ± 2.84% (*p* < 0.05) and 32.03 ± 10.51% (*p* < 0.01) for B c1, and 17.68 ± 5.8% (*p* < 0.0001) and 53.24 ± 8.49% (*p* < 0.0001) for B c2, respectively, which are significantly higher than that of 2.08 ± 0.23% and 3.31 ± 1.56% in the WT control (Figure 6C,D).

To test the mechanism through which SPAG5 impairs docetaxel-induced apoptosis, the mitochondrial membrane potential of MDA-MB-231 cells was measured using the TMRE-Mitochondrial Membrane Potential Assay Kit. Consistent with the above findings, the resistant WT control cells showed greater mitochondrial activity than knockout clones. A concentration of docetaxel at 1 nM reduced TMRE fluorescence by 26.23 ± 4.71% (*p* < 0.01) and 18.22 ± 2.18% (*p* < 0.05) in M c1 and M c2, respectively, compared to the WT control. A reduction of 31.95 ± 8.69% and 30.3 ± 0.31% was observed in M c1 and M c2 with 3 nM docetaxel treatment, respectively, when compared to the WT control (Appendix A).

These results imply that SPAG5 plays a major role in the cytotoxicity and apoptosis induced by doxorubicin and docetaxel. SPAG5 may mediate docetaxel-induced apoptosis through the mitochondrial pathway. Future research is warranted to explore the contribution of SPAG5 to the endoplasmic reticulum and death receptor pathways.

## 4. Discussion

The exact function of SPAG5 in TNBC remains controversial, as in the aforementioned data mining studies. SPAG5 may have varying degrees of impact on clinical outcomes in different breast cancer cohorts and combination chemotherapy. Therefore, an in-depth understanding of the oncogenic role of SPAG5 is essential to uncover TNBC cancer pathology and develop novel therapeutic strategies.

In cell line-based in vitro studies, a drastic reduction in cell proliferation and invasion was seen in SPAG5 knockdown TNBC cell lines [17,18,19]. SPAG5 might facilitate TNBC cell proliferation and invasion by activating the canonical WNT/β-catenin signalling pathway [18]. SPAG5 activates AKT/mTOR signalling and subsequently upregulates the expression and secretion of WNT3. The ligand binding of WNT3 to the receptor results in the dephosphorylation of β-catenin and the translocation of β-catenin into the nucleus. The dephosphorylated β-catenin interacts with T cell factor 4 (TCF4) and thus activates the gene transcription of oncogenes and epithelial-mesenchymal transition [13,18]. Furthermore, SPAG5 was reported to regulate the sensitivity of the TNBC cell lines (MDA-MB-231, MDA-MB-468, Hs578t and BT549) to the poly-ADP ribose polymerase (PARP) inhibitor Olaparib. The mechanism through which SPAG5 impairs Olaparib sensitivity might be by facilitating the c-MYC binding protein (MYCBP)/c-MYC axis, which upregulates homologous recombination (HR) DNA repair proteins and confers resistance to Olaparib by repairing the Olaparib-induced DNA lesions [17]. SPAG5 might also increase Olaparib resistance by promoting the cell cycle, especially the S/G2 transition. The decreased S-phase duration reduces the Olaparib-provoked DNA lesions, considering that DNA damage signal mainly arises in S-phase cells [26]. Although the above studies revealed the multiple roles of SPAG5 in TNBC cell behaviour, direct evidence for SPAG5-mediated chemoresistance is lacking, especially for the most widely used anthracycline- and taxane-based chemotherapy. Considering the limitations of the traditional gene knockdown techniques (e.g., incomplete protein depletion and high false positive rate), this study employed *SPAG5* gene knockout cell models to clarify the contribution of SPAG5 to the cytotoxicity of doxorubicin and docetaxel in the TNBC cells.

In the present study, *SPAG5* knockout by CRISPR-Cas9 in MDA-MB-231 and BT549 TNBC cells was confirmed on both the gene and protein levels. Two clones of each cell line were picked up to eliminate single-cell variability. All the selected clones were heterozygous, with a donor allele and an indel allele. Interestingly, the indel allele of different clones was with the same “T” insertion adjacent to *SPAG5* exon 1, irrespective of the gRNAs and cell lines. The mechanism of this phenomenon remains unknown. The DNA lesion induced by the Cas9 protein should be theoretically re-ligated by the NHEJ DNA repair route and end up with a random base pair insertion or deletion, which is considered a major characteristic of NHEJ. The present results may imply the involvement of other DNA repair pathways in the process of CRISPR-Cas9 gene editing in cancer cells. The locus of the “T” insertion interrupted *SPAG5* expression by damaging the GT-AG mRNA processing rule [25]. Thus, the HDR-mediated on-target integration of the donor template on the donor allele and the NHEJ-mediated “T” insertion on the indel allele collectively silenced the *SPAG5* expression. The complete deletion of the SPAG5 protein remarkably lowered the clonogenicity of *SPAG5* gene knockout clones compared to the WT control in both MDA-MB-231 and BT549 cells. These results were consistent with the previous gene knockdown-based studies on the SPAG5-regulated proliferation of the TNBC cells [17,18,19]. While the current study focused on the SPAG5-related chemoresistance, the decreased clonogenic capacity may be used as additional evidence of the functional deletion of SPAG5 in conjugation with the genomic PCR, DNA sequencing and Western blotting results.

The cytotoxicity assay demonstrated that both doxorubicin and docetaxel sensitivity for the knockout clones was increased by roughly 2-fold for both MDA-MB-231 and BT549 cells. The effect of the *SPAG5* knockout on doxorubicin- and docetaxel-induced apoptosis in MDA-MB-231 and BT549 cells was also tested. Deletion of *SPAG5* by CRISPR-Cas9 led to an increase in both doxorubicin- and docetaxel-induced caspase 3/7 activity in the TNBC cells. Deletion of *SPAG5* was associated with elevated apoptosis rate induced by docetaxel with downregulation of mitochondrial membrane potential. This demonstrated that SPAG5 may mediate docetaxel-induced apoptosis through the mitochondrial pathway. The above findings supported the current hypothesis that *SPAG5* gain and amplification is one of the major reasons for both doxorubicin and docetaxel resistance in the TNBC cells. Searching further for the related mechanisms, a recent study indicated that SPAG5 regulated the sensitivity to taxol (aka., paclitaxel), a standard chemotherapy for TNBC, in cervical cancer cells (HeLa and SiHa) through altering mTOR activity [5]. It was postulated that mTOR activation protected cervical cancer cells from apoptosis under taxol treatment [5]. Considering the similar mechanism of action between docetaxel and paclitaxel [27] and the distribution of mTOR at mitochondria and its role in maintaining mitochondrial membrane potential [28], the PI3K/AKT/mTOR signalling pathway may exert a vital effect on SPAG5-mediated docetaxel cytotoxicity in the TNBC cells. In the meantime, it was reported that SPAG5 prohibited DNA damage-induced apoptosis by downregulating p53 in cervical cancer cells [12] and upregulating c-MYC signalling in the TNBC cells [17]. The downregulated p53 suppressed the expression of BAX proapoptotic protein, and the upregulated c-MYC signalling promoted the expression of RAD51 and BRCA1/2 DNA repair proteins [12,17]. SPAG5 overexpression in the TNBC cells might facilitate these processes upon genotoxic stresses and prevent doxorubicin-induced apoptosis. Where the cytotoxicity of doxorubicin exploited DNA damage by intercalating into DNA and disrupting DNA repair, SPAG5 overexpression prevented the accumulation of DNA damage by altering p53 and c-MYC activity and influenced the eventual apoptosis. Further studies are required to better understand the exact molecular mechanisms by which SPAG5 is involved in the TNBC cells.

These results indicate that screening tumour SPAG5 expression has the potential to predict TNBC development and stratify patients for precision chemotherapy in the clinical setting. The lack of an effective way to understand patients’ response to chemotherapy always makes it challenging to select drugs. The TNBC patients taking advantage of SPAG5 stratification may contribute to the solution and improve their OS and survival quality. In addition, the current strategy to cope with chemoresistance is to apply combination chemotherapy or switch to other chemodrugs. The TNBC patients always end up with multi-drug resistance (MDR), and a cure is practically non-existent [29]. The findings in the present study depict that targeting SPAG5 may represent a novel strategy to control the development of TNBC and overcome chemoresistance. Future studies are encouraged to explore the predictive cutoff values, the mechanisms of SPAG5 in reversing TNBC chemoresistance and the drug-targeting potentials of SPAG5 in TNBC. Another question is whether SPAG5 enhances TNBC progression and hampers chemodrug sensitivity in vivo. 

This study suggested that SPAG5 antagonism might be a novel strategy for reversing chemoresistance in TNBC. However, the intracellular localisation and lack of enzymatic activity make it challenging to target SPAG5 and exploit catalytical activity. Many novel drug delivery approaches may contribute to overcoming these issues, such as cell-penetrating peptides (CPPs) [30,31] and nanoparticle carriers [32,33]. They are effective intracellular delivery vehicles for translocating various therapeutics, which enable the cargos to permeabilise and kill drug-resistant breast cancer cells. Alternatively, targeting the upstream regulators of SPAG5 rather than SPAG5 itself might contribute to attenuating SPAG5-induced oncogenic signalling. For example, emerging evidence suggested that the functional expression of SPAG5 is mediated by microRNAs (miRNAs). A previous study reported that the oncogenic functions of SPAG5 were suppressed by miR-10b-3p in TNBC cells [19]. In other cancer types, the SPAG5 expression and function were also correlated with miRNAs, such as miR-1179 in non-small cell lung cancer [34], miR-539 in prostate cancer [6] and miR-363-3p in hepatocellular cancer [14]. SPAG5 was also identified as a direct transcriptional target of the YAP/TAZ/TEAD axis in the TNBC cells. The depletion of the YAP/TAZ impaired SPAG5 expression and its oncogenic activity [19]. Targeting these SPAG5 regulators may be a potential therapeutic modality for cancer treatment. 

The limitations of the current study include the negative effects of CRISPR-Cas9 and the lack of validation in in vivo cancer models. In addition to the well-known drawbacks of CRISPR-Cas9, such as the endonuclease activity-induced cell damage and the off-target effect, it was reported that Cas9-induced DNA lesions might trigger *p53*- and *KRAS*-dependent DNA damage response and eventually cause cellular toxicity and cell death. This phenomenon might confer a selective advantage to the cells with *p53* and/or *KRAS* mutations [35,36,37]. CRIPSR-Cas9 has also been implicated in large-scale chromosome structural alterations, leading to potential ectopic expression of other genes [38,39]. But, CRISPR-Cas9 is still considered reliable and superior to other gene knockout and knockdown technologies. In addition, the present study failed to validate the SPAG5-mediated TNBC progression and chemoresistance in vivo xenograft tumour models, which mimic the human tumour microenvironment to guide the clinical application of SPAG5 [17,40].

## 5. Conclusions

In conclusion, the present study first shows that the recombinant plasmid-based CRISPR-Cas9 technology can be used to delete the SPAG5 gene in two TNBC cell lines. We discovered that targeting the intron–exon boundaries might also induce transcription failure and, thus, protein deletion, in addition to targeting exons. The same pattern of indel in different clones suggested the potential involvement of other DNA repair pathways in CRISPR-Cas9 gene editing. Additionally, the present study adds to the growing body of evidence that SPAG5 is associated with cancer progression and chemoresistance. We found that SPAG5 deletion compromised the proliferation and resistance to doxorubicin and docetaxel in the TNBC cells. SPAG5 might regulate docetaxel-induced apoptosis through the mitochondrial pathway. Based on these findings, we propose that SPAG5 is a promising biomarker in predicting response to chemotherapy and clinical outcomes. Further research on the predictive cutoff values of SPAG5, the mechanisms underlying SPAG5, and the development of anti-SPAG5 therapy is warranted.

## Figures and Tables

**Figure 1 cancers-16-01269-f001:**
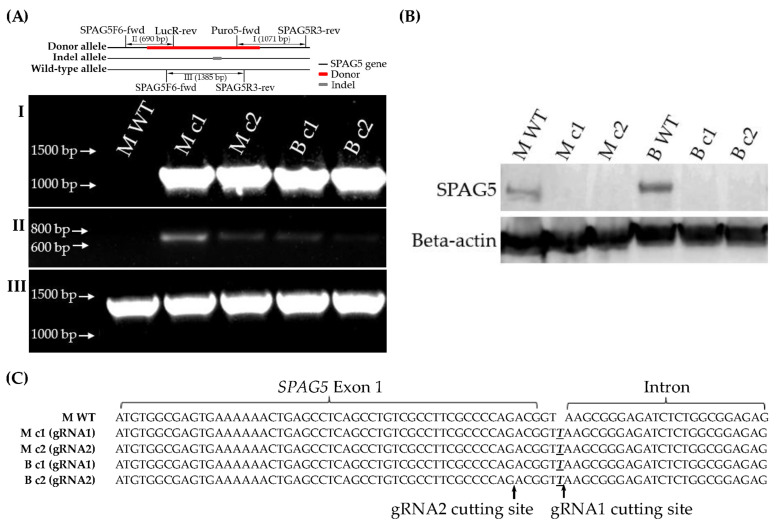
Establishment of *SPAG5* gene KO clones. The genomic DNA of single-cell colonies were PCR amplified using Puro5-fwd/SPAG5R3-rev (**A-I**) and SPAG5F6-fwd/LucR-rev (**A-II**) primers to examine the on-target integration of puromycin and luciferase gene on the donor allele, respectively. The SPAG5 protein expression was examined using western blotting (**B**). The indel allele was PCR amplified using SPAG5F6-fwd/SPAG5R3-rev primers (**A-III**), flanking the gRNA1/2 cutting sites and sequenced using Sanger sequencing (**C**). Emphasised bases were the inserted bases after nonhomologous end joining (NHEJ) repair. The uncropped blots are shown in the Appendix A.

**Figure 2 cancers-16-01269-f002:**
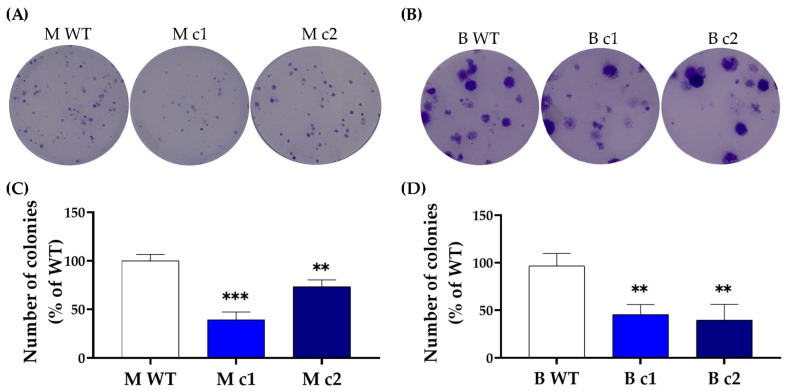
Representative clonogenic growth images of (**A**) MDA-MB-231 and (**B**) BT549 clones. MDA-MB-231 and BT549 clones at a density of 200 cells/well were seeded into a 6-well plate and incubated for 10 days. Colonies of more than 50 cells are scored for survival and counted using ImageJ software (1.50i). The number of colonies is presented as a mean percentage of WT control in (**C**) MDA-MB-231 and (**D**) BT549 cells. The bar represents the mean and standard deviation of three independent experiments performed in duplicates. **, *p* < 0.01; ***, *p* < 0.001 according to Dunnett’s post-hoc test that followed one-way ANOVA.

**Figure 3 cancers-16-01269-f003:**
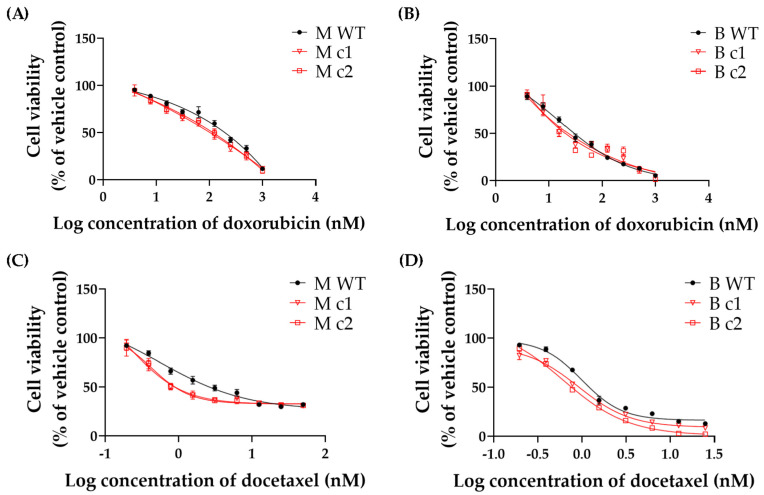
Representative (**A**,**B**) doxorubicin- and (**C**,**D**) docetaxel-induced growth inhibition in MDA-MB-231 and BT549 knockout clones, respectively. Doxorubicin and docetaxel were serially diluted threefold and twofold, respectively. Symbols are mean and standard deviation of triplicates. Solid lines are non-linear regression fits (Y = Bottom + (Top − Bottom)/(1 + 10^(LogIC50 − X)) to the data.

**Figure 4 cancers-16-01269-f004:**
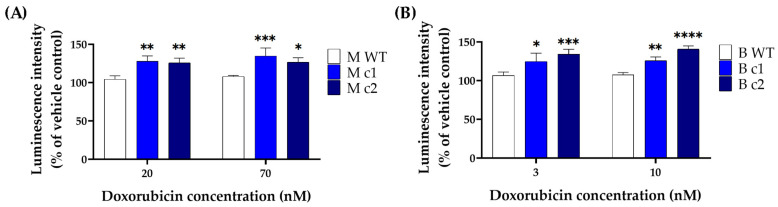
Caspase 3/7 activity in MDA-MB-231 (**A**) and BT549 (**B**) knockout clones treated with doxorubicin. MDA-MB-231 knockout clones were treated with 20 nM and 70 nM doxorubicin, and BT549 knockout clones were treated with 3 nM and 10 nM doxorubicin for 72 h, respectively. The caspase 3/7 activity was measured by Caspase-Glo^®^ 3/7 assay, and the luminescent signal was detected using a plate reader. Data are presented as a mean percentage of vehicle control. The bar represents the mean and standard deviation of three independent experiments performed in triplicates. *, *p* < 0.05; **, *p* < 0.01; ***, *p* < 0.001; ****, *p* < 0.0001 according to Sidak’s post-hoc test that followed two-way ANOVA.

**Figure 5 cancers-16-01269-f005:**
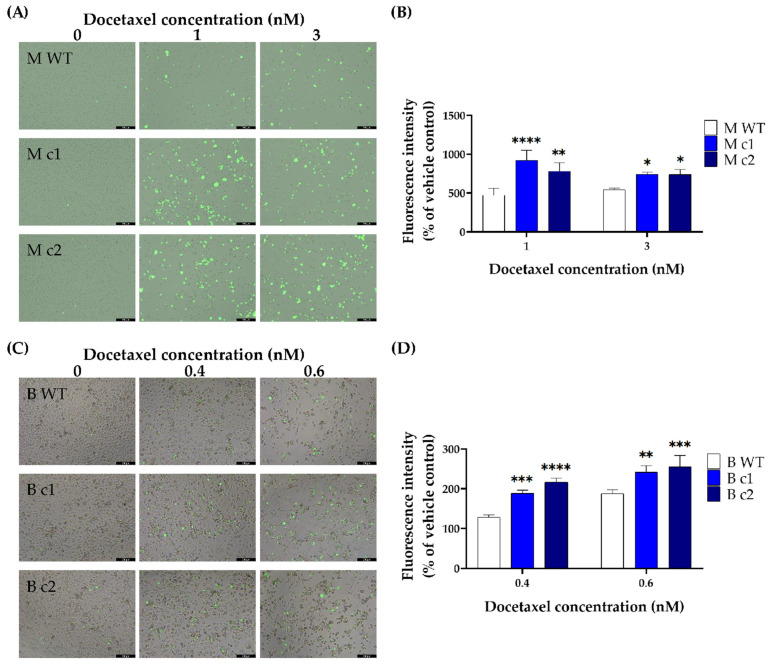
Representative docetaxel-induced caspase 3/7 activation in MDA-MB-231 (**A**,**B**) and BT549 (**C**,**D**) knockout clones. The clones were treated with different concentrations of docetaxel for 72 h. The apoptotic cells were stained with CellEvent^TM^ Caspase-3/7 Green Detection Reagent and visualised by fluorescence microscopy (scale bars, 200 μM). The fluorescence intensity was then measured using a plate reader. Data are presented as a mean percentage of vehicle control. The bar represents the mean and standard deviation of three independent experiments performed in triplicates. *, *p* < 0.05; **, *p* < 0.01; ***, *p* < 0.001; ****, *p* < 0.0001 according to Sidak’s post-hoc test that followed two-way ANOVA.

**Figure 6 cancers-16-01269-f006:**
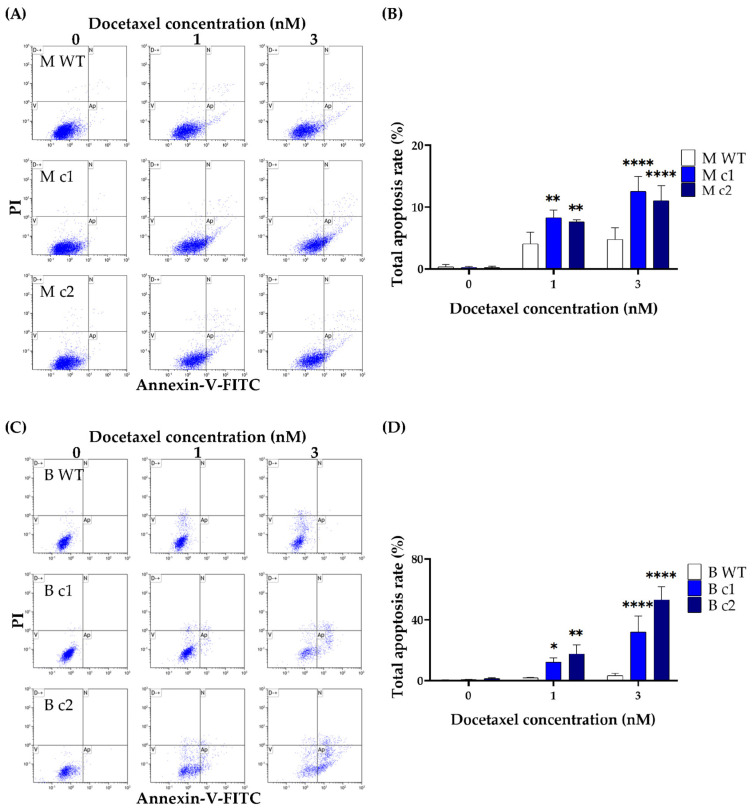
Docetaxel-induced apoptosis in MDA-MB-231 (**A**,**B**) and BT549 (**C**,**D**) knockout clones. The cells were treated with docetaxel (1 nM and 3 nM) for 72-h before being stained with Annexin V-FITC/PI, and the total apoptosis rate was measured using flow cytometry. The bar represents the mean and standard deviation of three independent experiments performed in duplicates. *, *p* < 0.05; **, *p* < 0.01; ****, *p* < 0.0001 according to Sidak’s post-hoc test that followed two-way ANOVA.

**Table 1 cancers-16-01269-t001:** Primer constructs used in PCR amplification.

Primer	Constructs
Puro 5	5′-TTGAATGGAAGGATGGAGCTAC-3′
SPAG5R3	5′-GACTGACCTTTCCGTAAGTGAC-3′
SPAG5F6	5′-GTTAGCCCAGTAAACTGGTAGC-3′
LucR	5′-GTCTTCGAGTGGGTAGAATGGC-3′

**Table 2 cancers-16-01269-t002:** Doxorubicin- and docetaxel-induced growth inhibition in MDA-MB-231 knockout clones: Comparison of IC_50_ values between WT control and knockout clones. Values are the mean and standard deviation of three independent experiments performed in triplicates. *p*-values according to Dunnett’s post-hoc test that followed one-way ANOVA.

A, Doxorubicin		
Cell type	IC_50_, nM	*p*-value
M WT	199.667 ± 47.919	
M c1	81.79 ± 15.342	0.0046
M c2	103.967 ± 2.909	0.0123
B, Docetaxel		
Cell type	IC_50_, nM	*p*-value
M WT	1.009 ± 0.062	
M c1	0.488 ± 0.045	0.0004
M c2	0.557 ± 0.117	0.0008

**Table 3 cancers-16-01269-t003:** Doxorubicin- and docetaxel-induced growth inhibition in BT549 knockout clones: Comparison of IC_50_ values between WT control and knockout clones. Values are the mean and standard deviation of three independent experiments performed in triplicates. *p*-values according to Dunnett’s post-hoc test that followed one-way ANOVA.

A, Doxorubicin		
Cell type	IC_50_, nM	*p*-value
B WT	30.657 ± 5.512	
B c1	19.113 ± 2.048	0.0446
B c2	8.465 ± 5.847	0.0023
B, Docetaxel		
Cell type	IC_50_, nM	*p*-value
B WT	0.905 ± 0.107	
B c1	0.676 ± 0.116	0.0142
B c2	0.658 ± 0.112	0.0106

## Data Availability

The datasets used and/or analysed during the current study are available from the corresponding author upon reasonable request.

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
