# Peer review of "Sperm-Associated Antigen 5 Knockout Reduces Doxorubicin and Docetaxel Resistance in Triple-Negative Breast Cancer MDA-MB-231 and BT549 Cells"

_cancers, 2024, doi:10.3390/cancers16071269_

Round 1

Reviewer 1 Report

Comments and Suggestions for Authors

Dear Authors,

I congratulate you for the interesting work.

The paper is well written and organized, and I think that it can be accepted in present form.

Sincerely,

Author Response

Thank you for your comments on our manuscript.

Reviewer 2 Report

Comments and Suggestions for Authors

The authors studied the mechanism of resistance to doxorubicin and paclitaxel in human TNBC cell lines. Triple negative breast cancer is the most difficult to treat breast cancer subtype and there is no targeted therapy for these patients. They conducted their study on Sperm-associated antigen 5 (SPAG5) which is biomarker for cellular resistance to breast cancer therapies. 

They performed the knockout of SPAG5 in two human TNBC cell lines and these cell lines were used for further tests of sensitivity to doxorubicin and paclitaxel. They showed in MTT test that doxorubicin and paclitaxel induced the growth inhibition and increased the apoptosis in two cell lines with the knockout of SPAG5. The manuscript is interesting and raises important issue of resistance to chemotherapy. 

Author Response

(The authors gave the same response as above.)

Reviewer 3 Report

Comments and Suggestions for Authors

First of all, I would like to thank you for inviting me to review the manuscript entitled: ’Sperm-Associated Antigen 5 Knockout Reduces Doxorubicin and Docetaxel Resistance in Triple-Negative Breast Cancer MDA-MB-231 and BT549 Cells’. The general conclusion demonstrated that recombinant plasmid-based CRISPR-Cas9 technology can be used to delete the SPAG5 gene in TNBC cell lines. SPAG5 has an important role in regulating cell proliferation and doxorubicin- and docetaxel-resistance in MDA-MB-231 and BT549 cells. I recommend publication after some major issues have been clarified:

Major:
1. What is the novelty of this study? and how your outcomes can help further studies in this area. Please add a brief explanation of the novelty of this study in the introduction section.
2. Please provide explanation, how we can applicate your results into practice?, why your work is valuable in the field?
3. Based on such a large number of results, the authors can expand the conclusions.
4. Based on the reference list, the novelty of the study can be doubted, as only 11 (27.5%) (11 out of 40) references are from the last 5 years, while 16 (40%) references are published before 2015.
5. Discussion section: Please use updated papers to discuss and compare your results with the available literature.

Minor:
1. Fonts in almost all figures need to be larger, some labels are not visible at all.
2. Please remove the dots between A and B and C and D at the top of all figures
3. Please provide limitations of the study.

General: interesting, well-conducted work.

Reviewer 4 Report

Comments and Suggestions for Authors

Athors: Ji He, Jiawei Li, Yanbiao Liu, Yan Li  

Title: Sperm-Associated Antigen 5 Knockout Reduces Doxorubicin and Docetaxel Resistance in Triple-Negative Breast Cancer MDA-MB-231 and BT549 Cells 

Comments: 

The Authors have performed a good study dedicated to elucidation of a role of SPAG5 in TNBC and its resistance to chemotherapy. Using CRISPR-Cas-9 technology they arranged SPAG5 knockdown in two TNBC-derived cell lines. Those cells (with SPAG5 knockdown) exhibited lower proliferation and impaired resistance to doxorubicin and docetaxel. The obtained results characterize SPAG5 as a promising molecular target to combat TNBC. The study has been performed at the high molecular level; the manuscrit is well written and nicely illustrated. The present material will be interesting for molecular oncologists, molecular pharmacologists and chemotherapists. 

Author Response

(The authors gave the same response as above.)

Round 2

Reviewer 3 Report

Comments and Suggestions for Authors

The authors have incorporated all the suggested modifications, and on this basis I suggest accepting the manuscript in its current form.